# Cu Nanoparticles Modified Step-Scheme Cu_2_O/WO_3_ Heterojunction Nanoflakes for Visible-Light-Driven Conversion of CO_2_ to CH_4_

**DOI:** 10.3390/nano12132284

**Published:** 2022-07-02

**Authors:** Weina Shi, Ji-Chao Wang, Aimin Chen, Xin Xu, Shuai Wang, Renlong Li, Wanqing Zhang, Yuxia Hou

**Affiliations:** 1School of Chemistry and Materials Engineering, Xinxiang University, Xinxiang 453000, China; shiweina516@163.com (W.S.); chenaimin1978@163.com (A.C.); xuxin202111@163.com (X.X.); wangshuai00129@163.com (S.W.); 2College of Chemistry and Chemical Engineering, Henan Institute of Science and Technology, Xinxiang 453000, China; rlli@hist.edu.cn (R.L.); zhangwqzzu@163.com (W.Z.); 3College of Chemistry, Zhengzhou University, Zhengzhou 450000, China

**Keywords:** CO_2_ reduction, selectivity, Cu/Cu_2_O/WO_3_, photocatalysis, S-scheme

## Abstract

In this study, Cu and Cu_2_O hybrid nanoparticles were synthesized onto the WO_3_ nanoflake film using a one-step electrodeposition method. The critical advance is the use of a heterojunction consisting of WO_3_ flakes and Cu_2_O as an innovative stack design, thereby achieving excellent performance for CO_2_ photoreduction with water vapor under visible light irradiation. Notably, with the modified Cu nanoparticles, the selectivity of CH_4_ increased from nearly 0% to 96.7%, while that of CO fell down from 94.5% to 0%. The yields of CH_4_, H_2_ and O_2_ reached 2.43, 0.32 and 3.45 mmol/g_cat_ after 24 h of visible light irradiation, respectively. The boosted photocatalytic performance primarily originated from effective charge-transfer in the heterojunction and acceleration of electron-proton transfer in the presence of Cu nanoparticles. The S-scheme charge transfer mode was further proposed by the in situ-XPS measurement. In this regard, the heterojunction construction showed great significance in the design of efficient catalysts for CO_2_ photoreduction application.

## 1. Introduction

The rapid growth of atmospheric carbon dioxide (CO_2_) concentration has attracted considerable attention due to its greenhouse effect on the global climate [1,2]. Developing artificial photosynthesis routes by reducing anthropogenic CO_2_ emissions using solar energy instead of carbon-based fuels such as methane, methanol or carbon monoxide is still an intensive research topic in the environmental and energy field [2,3,4]. Since a TiO_2_ photocatalyst was used to reduce CO_2_ into methanol and formaldehyde by Inoue et al. in 1972 [5], most semiconductor photocatalysts, such as g-C_3_N_4_, CeO_2_, W_18_O_49_ and Bi_2_WO_6_ have received immense attention for CO_2_ reduction under visible light irradiation, which occupies 44% of solar light [6,7,8,9,10,11]. Among the common photocatalysts, WO_3_ material has gradually gained special scientific interest due to its relatively narrow bandgap (E_g_~2.8 eV) and unique crystal structure containing a network of corner-shared octahedral units of [WO_6_], which could theoretically utilize 12% of solar light and enhance charge migration in the catalyst [12]. Although WO_3_ crystals exhibit various phases, few studies have focused on hexagonal phase WO_3_ (*h*-WO_3_) for photocatalytic CO_2_ reduction [8,13,14]. *h*-WO_3_ not only exhibits excellent visible light responsive properties according to its good photochromic ability [15], but also strong CO_2_ adsorption at low-pressure due to the presence of ultramicro-sized tunnels [16]. It is thus considered as a potential visible-light-driven (VLD) photocatalyst for CO_2_ reduction. However, the application of photocatalytic CO_2_ reduction for *h*-WO_3_ is seriously hindered by poor charge separation and reduction ability [17,18].

It is well known that two-dimensional (2D) semiconductors exhibit superior solar-driven photocatalytic activity as a result of improved photoexcited charge separation, compared with that of bulk photocatalysts [19,20]. The 2D nanostructure also greatly influences the optical properties and electronic properties of semiconductors [21]. In particular, the travel distance of photoexcited carriers in the WO_3_ nanoflake becomes short, and more photons can be adsorbed by the nanoflake in a remarkably short time under low photon flux density due to its large surface area [22]. It is thus considered that *h*-WO_3_ nanoflakes could reduce charge recombination and improve the performance of photocatalytic CO_2_ reduction. In addition, there have been few reports on the hydrothermal fabrication of nanostructured *h*-WO_3_ photocatalysts [16,23,24], and the metastable hexagonal phase has limited its development and application [17,25]. Therefore, it is still a conceptual challenge in the field of materials research to fabricate heat-resisting *h*-WO_3_ nanoflakes for CO_2_ reduction. Apart from nanostructure engineering, heterojunction construction is also a prospective way of achieving improved redox abilities and efficient separation of photoexcited electrons and holes for *h*-WO_3_ [2,26]. Among the numerous heterojunction photocatalysts, the heterostructured system with a staggered band alignment has drawn much attention due to its efficient charge separation [27]. Notably, the charge transfer (step-scheme mode) in the above junction directly quenches the weaker oxidative holes and reductive electrons, which is preferable to obtain photoexcited carriers with stronger redox abilities [28]. Recently, cuprous oxide (Cu_2_O) with a conduction band of −1.15 eV (vs. NHE) has been utilized as a VLD semiconductor photocatalyst or co-catalyst for CO_2_ reduction [29,30]. Nevertheless, Cu_2_O exhibits weak oxidation ability and poor photostability. An effective way to address these issues may be coupling Cu_2_O and WO_3_ to construct a heterostructured system with a step-scheme (S-scheme) charge transportation mode. S-scheme systems consisting of monoclinic or orthorhombic WO_3_ and Cu_2_O were constructed for organic degradation, CO_2_ reduction or photoelectrocatalytic water splitting [31,32,33]. To our best knowledge, there are few studies regarding hexagonal phase WO_3_ nanoflakes and Cu_2_O for photocatalytic CO_2_ reduction under visible light irradiation.

Except for enhanced photocatalytic activity for CO_2_ reduction, the selectivity of multiple hydrocarbon products plays a key role in their further application in the chemical industry. The influence of the specific surface structure of the photocatalyst cannot be ignored in the process of CO_2_ reduction [34,35]. Due to the eight-electron reaction of CH_4_ generation from CO_2_, loading co-catalysts with fast electron transfer, including Au, Pt, Ag and Ti, could be conductive to control the reduction of products [35,36,37,38]. Non-noble-metal Cu co-catalyst has become a newly emerging research spot for CO_2_ reduction. Albo et al. reported the deposition of Cu nanoparticles on the TiO_2_ surface, which facilitated the photocatalytic process from CO_2_ to CH_3_OH [39]. Meng et al. proved that Cu co-catalyst played a pivotal role in increasing methane selectivity in the CO_2_ photoreduction process for the S doped g-C_3_N_4_ catalyst [40]. In the Cu_2_O/WO_3_ system, loading Cu co-catalysts can originate from Cu_2_O with unsaturated coordination sites dispersed on the surface of the catalyst, and offer more catalytic centers and act as the electron sink to increase the concentration of charge carriers. Therefore, the construction of a Cu/Cu_2_O/WO_3_ catalyst could accelerate charge separation across the Cu_2_O/WO_3_ heterojunction interface and modulate product selectivity for photocatalytic CO_2_ reduction.

Herein, we report the synthesis of Cu and Cu_2_O species onto WO_3_ nanoflakes by a one-step electrodeposition method. The photocatalytic performances for CO_2_ reduction with water vapor over the obtained samples were investigated under visible light irradiation (λ > 400 nm), and the influence of Cu nanoparticles on product selectivity was studied. The band structure of the heterojunction was measured and the S-scheme charge-transfer mode was further verified.

## 2. Materials and Methods

### 2.1. Materials Synthesis

WO_3_ nanoflake film was synthesized by the solvothermal-calcination method. Before solvothermal growth, a thin seed layer was deposited onto the fluorine-doped tin oxide (FTO)-coated glass substrate (2.3 cm × 2.0 cm) by spin coating the precursor solution, which was made by dissolving 1.25 g of H_2_WO_4_ in 30 wt% H_2_O_2_ (8 mL), followed by annealing at 500 °C for 2 h in air. The H_2_WO_4_ solution for solvothermal treatment was prepared by dissolving 1.25 g of H_2_WO_4_ into 30 wt% H_2_O_2_ (15 mL) and heating at 95 °C for 2 h. Nanoflake growth was achieved using 3 mL of H_2_WO_4_ solution mixed with 0.5 mL of HCl (6 mol/L) and 12.5 mL of acetonitrile. A vertically oriented FTO-glass substrate was immersed into the above solution and placed within a Teflon lined stainless steel autoclave, which was then sealed and maintained at 180 °C for 3 h. The substrate was then rinsed with deionized water and dried in air. The WO_3_ nanoflake film was obtained after annealing at 500 °C for 2 h in air.

The electrodeposition was performed in a standard three-electrode system. The FTO substrates (2.3 cm × 2.0 cm) were used as the working electrodes. The platinum sheet (1.0 cm × 1.0 cm × 0.2 cm) and Ag/AgCl electrode were used as the counter and reference electrodes, respectively. All of the chemicals were used without further purification. Before electrodeposition, the substrates were rinsed with distilled water and then cleaned by ultrasonic treatment in ethanol for 10 min. Cu_2_O was electrodeposited in 0.02 mol/L Cu(OAc)_2_ aqueous solution (50 mL) containing 0.02 mol/L acetic acid (5 mL) using chronopotentiometry at −0.4 V with 0.5 s for 150 cycles. The Cu/Cu_2_O sample was obtained at −0.7 V under the similar conditions as above.

Cu_2_O/WO_3_ and Cu/Cu_2_O/WO_3_ films were synthesized by electrodeposition of Cu_2_O and Cu/Cu_2_O on the obtained WO_3_ nanoflakes, respectively. Typically, the deposition process was conducted in Cu(OAc)_2_ aqueous solution (0.02 mol/L, 50 mL) using chronopotentiometry electrodeposition with 0.5 s in 150 cycles, and the Cu_2_O/WO_3_ and Cu/Cu_2_O/WO_3_ films samples were obtained at −0.4 and −0.7 V, respectively. All the samples were then rinsed with deionized water and dried in air. Additionally, Cu_2_O and Cu/Cu_2_O films were prepared by the similar method, and Cu_2_O and Cu/Cu_2_O nanoparticles were broken off from the above samples by the ultrasound method. The dispersion solutions of Cu_2_O and Cu/Cu_2_O nanoparticles were then severally sprayed onto the WO_3_ nanoflakes, and the obtained films were finally denoted as Cu_2_O/WO_3_ m and Cu/Cu_2_O/WO_3_ m samples.

### 2.2. Characterization

The crystal phases of the samples were recorded using an X−ray diffractometer (PANalytical X’ pert PRO, Netherlands) with a Cu Kα irradiation source (λ = 0.154 nm) and 0.15°/s scanning step. A scanning electron microscope (SEM, Nova NanoSEM 450, FEI) using the acceleration 300 kV voltage was used to characterize the morphology of the obtained products. Transmission electron microscopy (TEM) was obtained on a Tecnai G2 F20 S-TWIN electron microscope. Furthermore, high-resolution transmission electron microscopy (HRTEM) and Energy Dispersive X-ray spectroscopy (EDX) was employed and the corresponding fast Fourier transform (FFT) was evaluated by Gatan Digital Micro-graph software (Gatan Inc., Pleasanton, CA, USA). X-ray photoelectron spectroscopy (XPS) measurements were carried out at room temperature on a Thermo escalab 250Xi X-ray Photoelectron Spectrometer with a monochromatic Al Kα radiation (h*v* = 1486.6 eV). For XPS analysis, the samples without exposure to air were dried in N_2_ flow gas and vacuum packed to avoid any impurity. All spectra were calibrated to the C_1s_ peak at 284.6 eV. The peak position was estimated using a fitting procedure based on the summation of Lorentzian and Gaussian functions using the XPSPEAK 4.1 program. UV-Vis diffuse reflectance spectra (DRS) were performed on a scan UV-Vis spectrometer (Cary 5000). The composition for the composites was determined by ICP-AES analysis using Thermo Scientic iCAP 6000 spectrometry. The photoelectrochemical test was recorded in a conventional three-electrode system by a CHI 660E electrochemical workstation (Chenhua, Shanghai, China). The photocurrents of the photocatalysts were measured at 0.0 V (vs. Ag/AgCl) in Na_2_SO_4_ aqueous solution after being purged by N_2_ under UV-visible light with an AM 1.5 G filter.

### 2.3. Photocatalytic Performance Tests

Photocatalytic CO_2_ reduction activity with gaseous phase H_2_O was evaluated in a CEL-HPR100 stainless steel cylindrical vessel (Beijing China Education Au-Light Co., Ltd., China), and the light source was a PLS-SXE 300 Xenon arc lamp with a UV cutoff filter (λ > 400 nm). The compressed high purity CO_2_ gas (99.995%) was passed across a deionized water bubbler, which generated a mixture of CO_2_ and H_2_O vapor. After visible-light irradiation, the product yields and types in the gas phase were analyzed using a GC-7890II gas chromatograph (Beijing China Education Au-Light Co., Ltd., China), and the hydrocarbon product was further analyzed by 7890A-5975C GC-MS (Agilent Technologies Inc., Santa Clara, CA, USA). The experimental process is detailed in Appendix A.

The product selectivity (S) for CO_2_ reduction was calculated with the following Equations (1)–(3):S_CO_ (%) = 2 × N_CO_/(2 × N_CO_ + 8 × N_CH4_ + 2 × N_H2_) × 100(1)
S_H2_ (%) = 2 × N_H2_/(2 × N_CO_ + 8 × N_CH4_ + 2 × N_H2_) × 100(2)
S_CH4_ (%) = 2 × N_CH4_/(2 × N_CO_ + 8 × N_CH4_ + 2 × N_H2_) × 100(3)

N_CO_, N_CH4_ and N_H2_ were on behalf of the yield of detected CO, CH_4_ and H_2_ molecules in the photocatalytic process of CO_2_ reduction with H_2_O vapor.

## 3. Results and Discussion

### 3.1. Structure, Composition and Morphology

The XRD patterns of the as-prepared samples are presented in Figure 1. Several diffraction peaks at 13.9, 23.2, 27.2, 28.1, 47.4 and 49.8°(marked with black dotted lines) were observed corresponding to (1 0 0), (0 0 2), (1 0 2), (2 0 0), (0 0 4) and (2 2 0) planes of hexagonal WO_3_ (PDF card No. 01-085-2460). This is because the percentage of exposed facets was estimated by the respective peak areas of the facets [41]. Hence, the WO_3_ component in the bare and composite samples preferentially exposed (0 0 2) facets. Simultaneously, the diffraction peaks of Cu_2_O sample (marked with green dotted lines) matched perfectly with those of cubic phase Cu_2_O (PDF card No. 01-078-2076). To further explore the existence of Cu metal in the composite, the XRD pattern of Cu/Cu_2_O sample was studied, avoiding interference from the diffraction peak of hexagonal WO_3_, and the characteristic peaks at 43.4 and 50.6° (marked with pink dotted lines) were attributed to metallic Cu (PDF card No. 03-065-9743). Moreover, Cu metal was proven to be present in the Cu/Cu_2_O/WO_3_ composite. As a result, Cu and Cu_2_O were simultaneously electrodeposited onto hexagonal WO_3_, indicating the successful construction of the Cu/Cu_2_O/WO_3_ composite.

X-ray photoelectron spectroscopy (XPS) measurements were carried out to elucidate the surface composition and chemical states of the elements. The survey XPS spectrum of the typical Cu/Cu_2_O/WO_3_ sample indicated that the composite mainly consisted of W, Cu and O electrons. To gain further insight into the chemical bonding between W and other atoms in the composite, the high resolution XPS spectrum of W 4f (Figure 2a) was deconvoluted by Gaussian-Lorenzian analysis. The peaks at binding energies of 37.7 and 35.7 eV were ascribed to W (VI) state in tungsten oxide materials, while two distinct peaks at 34.5 and 36.4 eV were consistent with the values of W (V) oxidation state for all the samples [42,43]. The existence of W (V) was necessary to maintain the opening structure of hexagonal WO_3_ [14]. However, the area ratios of W (V) and W (VI) in the composite were higher than those of the bare WO_3_ sample (Appendix A) due to the electroreduction process for Cu_2_O deposition. In the high-resolution Cu 2p XPS spectrum (Figure 2b), two conspicuous peaks were observed at binding energies of 952.3 eV for Cu (I) 2p_1/2_ and 932.5 eV for Cu(I) 2p_3/2_ [44]. Furthermore, the Cu and W diffraction peaks of the composite shifted slightly compared with those of pure samples, which was the reason that the intense interaction existed between WO_3_ and Cu_2_O component in the composites, implying the formation of heterojunction. Cu LMM Auger spectra (Figure 2c) were further performed to explore the chemical state of the Cu element, The Auger parameter can be calculated from the equation of α′ = *E*_k_ (Auger electron) + *E*_b_ (photoelectron). Here, *E*_k_ is the kinetic energy, and *E*_b_ is the binding energy. The Auger parameter values of the Cu (I) and Cu (0) in the Cu/Cu_2_O/WO_3_ composite were determined to be 1848.7 and 1851.2 eV, respectively, which indicated the existence of Cu (I) and Cu (0) in the Cu/Cu_2_O/WO_3_ sample [45,46].

The morphologies of the as-obtained WO_3_, Cu_2_O/WO_3_ and Cu/Cu_2_O/WO_3_ powders were visualized using SEM images. As shown in Figure 3, all the samples exhibited the typical sheet morphology with various sizes. Furthermore, the SEM image in Figure 3a showed that the nanoflake surface was rough and porous. As shown in Figure 3b,c, both of the composites kept uniform sheet morphology, and the sheet thicknesses after electro-deposition of Cu_2_O and Cu/Cu_2_O had no obvious changes compared with that of bare WO_3_. Although there was no obvious difference in Cu content between Cu_2_O/WO_3_ and Cu/Cu_2_O/WO_3_ according to the ICP-AES analysis (Appendix A), the nanoflake surface of Cu/Cu_2_O/WO_3_ exhibited distinct embossment (Figure 3c).

Micro-structures of the obtained samples were investigated by TEM observation. Figure 4 revealed that the typical nanoflake-like morphology was observed with a sharp edge and clean boundary. In Figure 4a, the WO_3_ nanoflake had a porous surface morphology. The fringe spacings of 0.634 and 0.366 nm were discovered (inset of Figure 4a), which were consistent with the (1 0 0) and (0 0 2) lattice planes of hexagonal WO_3_, respectively. In Figure 4b, Cu_2_O/WO_3_ still kept a nanoflake-like morphology with smaller porous sizes compared with that of WO_3_. The lattice fringes with spacings of 0.246 and 0.213 nm were distinctly found (inset of Figure 4b) corresponding to the (1 1 1) and (2 0 0) planes of cubic Cu_2_O. Similar nanoflake-like morphology was observed for the Cu/Cu_2_O/WO_3_ sample in Figure 4c, and the (2 0 0) crystal facet of metallic Cu with lattice spacing of 0.180 nm was observed, except for the crystal facet of Cu_2_O (top-right inset of Figure 4c). Furthermore, the scattered Cu nanoparticles around the Cu/Cu_2_O/WO_3_ nanoflake by ultrasonic exfoliation were observed with clear lattice fringes of 0.180 and 0.208 nm were observed around the nanoflake (bottom-left inset of Figure 4c), and the size distribution of Cu nanoparticles in the selected area (framed in red, corresponding enlarged view in Appendix A) obeyed a logical normal distribution with an average diameter of 5.6 ± 1.1 nm (Appendix A). According to the elemental mapping images in Figure 4d, W, Cu and O elements were uniformly distributed in the Cu/Cu_2_O/WO_3_ sample, and the Cu element appeared on the surface, as well as around the nanoflakes. The above results demonstrate that Cu_2_O was uniformly electrodeposited on the surface of the WO_3_ nanoflake, and metallic Cu also existed in the Cu/Cu_2_O/WO_3_ sample.

The light absorption property is crucial for the utilization efficiency of solar energy for the catalyst. According to the UV-Vis diffuse reflectance spectra (Figure 5), all the composites showed strong absorption in the visible light region, indicating the feasibility of utilizing visible light for CO_2_ photoreduction. The band gaps (*E*_g_) of the samples were estimated by the plots of (α*hν*)^2^ versus photo energy (*hν*) (Appendix A), and the calculated *E*_g_ values of WO_3_ and Cu_2_O were 2.86 and 2.05 eV, respectively.

### 3.2. Photocatalytic Performance of CO_2_ Reduction

The photocatalytic CO_2_ reduction experiments were carried out under visible-light irradiation (> 400 nm) for the as-prepared catalysts. The photocatalytic performances of CO_2_ reduction with water vapor in Figure 6a showed that no products were generated over the WO_3_, Cu_2_O and Cu/Cu_2_O samples. Conversely, CO, CH_4_, O_2_ and H_2_ products were obtained for the Cu/Cu_2_O/WO_3_ and Cu_2_O/WO_3_ composites, and no other products such as HCHO or CH_3_OH were detected by either GC or GC-MS analyses. Furthermore, the product yields decreased for the mechanically dispersed Cu_2_O/WO_3_ m sample compared with those of the Cu_2_O/WO_3_, indicating that the loosely contacted interface was insufficient for photocatalytic CO_2_ reduction. Additionally, the product selectivity for CO_2_ reduction was analyzed for the Cu/Cu_2_O/WO_3_ and Cu_2_O/WO_3_ catalyst. Typically, the S_CH4_ and S_CO_ values of Cu_2_O/WO_3_ were calculated to be about 0.1% and 94.5%, which demonstrated that CO_2_ reactant was thermodynamically favorable to form CO by the two-electron reduction pathway. However, the Sc_H4_ and S_CO_ values of Cu/Cu_2_O/WO_3_ were calculated to be 96.7% and 0.0%, indicating that the metallic Cu promoted the formation and utilization of proton-assisted multi-electrons pathway in the photocatalytic CO_2_ reduction process [47]. The CH_4_, CO and O_2_ yields for CO_2_ photoreduction reached 1.87, 0.0065 and 2.63 mmol/g_cat_, respectively. Compared with the photocatalytic activity of the reported catalysts (Appendix A), the maximum rate of CH_4_ product over Cu/Cu_2_O/WO_3_ was obviously higher and high product selectivity was also obtained. To explore the influence of loading Cu on the photocatalytic performance, the controlled experiments in different atmospheres were further investigated (Figure 6b). In the CO/H_2_O atmosphere, more CH_4_ molecules were produced over Cu/Cu_2_O/WO_3_ compared with that of Cu_2_O/WO_3_. Meanwhile, in the N_2_/H_2_O atmosphere, the H_2_ yield from water splitting was improved with the existence of Cu in the composite, which indicated that Cu nanoparticles promoted the electron transfer and proton aggregation on the catalyst surface, improving the hydrogenated process for CO_2_ reduction over the Cu/Cu_2_O/WO_3_ catalyst. The yields of products for the Cu/Cu_2_O/WO_3_ composite were further investigated in the CO_2_/H_2_O atmosphere within 24 h of irradiation. As shown in Figure 6c, the total yields of CH_4_, H_2_ and O_2_ were enhanced with prolonging the irradiation time. The maximal yield rate of H_2_ reached 0.013 mmol/g_cat_/h at 24 h, and those of CH_4_ and O_2_ at 18 h were found to be 0.104 and 0.147 mmol/g_cat_/h, respectively. The decreased yields of the photocatalytic CO_2_ reduction were possibly due to oxidation of the formed carbonous compounds on the photocatalyst surface or the coverage of active sites by intermediates. Reproducibility and durability are critical for the long-term use of a catalyst in practical application. The results in Figure 6d show that the yields of CH_4_ and H_2_ for the typical Cu/Cu_2_O/WO_3_ catalyst slightly decreased, and the O_2_ yield obviously fell down after 5 cycles. Notably, after the 5th cycling experiment, the photocatalyst was regenerated by the electro-reduction method at 0.05 V in 0.5 M Na_2_SO_4_ solution. In the 6th cycling experiment, all the yields were obviously promoted, which were similar to those of the fresh catalyst. To illustrate the stability of the Cu/Cu_2_O/WO_3_ catalyst, XPS measurements were conducted to investigate the change in chemical composition after the 5th cycle. As shown in the Cu 2p and W 4f XPS spectra (Appendix A), Cu (II) and W (V) were identified, which were probably derived from the oxidation of Cu (I) and reduction in W (VI), and W (V) possibly became the high recombination center of the photogenerated electrons and holes, leading to a slight decrease in the photocatalytic activity for CO_2_ reduction.

### 3.3. Possible Photocatalytic Mechanism

Transient photocurrent response and photoluminescence spectra were measured to investigate the charge separation. Based on the photo-electric properties during several on–off illumination cycles (Figure 7a), the photocurrent density of Cu_2_O/WO_3_ was higher than those of the bare WO_3_ and Cu_2_O samples, and the Cu/Cu_2_O/WO_3_ sample exhibited the highest photo-induced density among them. Moreover, the photoluminescence spectra of the photocatalysts at the excitation wavelength of 405 nm are present in Figure 7b, and the emission intensity of the Cu/Cu_2_O/WO_3_ composite distinctly decreased compared with those of the WO_3_, Cu_2_O and Cu_2_O/WO_3_ samples. The above results indicate the promoted separation of photogenerated electron-hole pairs in the Cu/Cu_2_O/WO_3_ composite.

The band edge position and charge transport mode in the composite directly influenced the separation and redox ability of photoexcited charge carriers. Ultraviolet photoelectron spectra combined with *E*_g_ analysis were used to determine the electronic band structure of the catalyst. Figure 8a shows the UPS spectra of Cu_2_O and WO_3_. On the basis of the linear intersection method [48], the valence band (VB) of WO_3_ was estimated to be −7.33 eV (vs. vacuum), and the conduction band (CB) was −4.47 eV (vs. vacuum) based on the *E*_g_ value of WO_3_. According to the connection between vacuum energy and normal electrode potential (NHE) [29], the corresponding VB and CB positions of WO_3_ were 2.89 and 0.03 eV (vs. NHE), respectively. Similarly, the VB and CB values of Cu_2_O were separately calculated to be 0.91 and −1.14 eV (vs. NHE), which agreed well with previously reported results [31,49]. Hence, the heterostructure with staggered band alignment was successfully formed in this Cu_2_O/WO_3_ system, assuming that the possible band bending of the semiconductor was neglected. Based on the band energy structure of WO_3_ and Cu_2_O, two possible mechanisms for photo-induced carriers were proposed and described in Figure 8b,c. If the photo-excited charge carriers transferred according to the traditional model (Figure 8b), the photo-excited holes in the VB of WO_3_ would transfer to the VB of Cu_2_O, and the photo-excited electrons in the CB of Cu_2_O would migrate to the CB of WO_3_, where the accumulated electrons would not reduce CO_2_ to CO/CH_4_ or produce H_2_ from H_2_O on account of the more positive CB edge potential (0.21 eV vs. NHE) than the standard CO_2_ reduction potential [4]. Similarly, the accumulated holes in Cu_2_O would not accomplish H_2_O oxidation due to the more negative VB of Cu_2_O. Hence, the hypothesis of the traditional double-transfer model was invalid. The S-scheme charge transfer mode (Figure 8c) was proposed according to the enhanced photocatalytic activity of Cu/Cu_2_O/WO_3_. To verify the S-scheme charge transfer mode in this heterojunction, the XPS spectra of Cu_2_O/WO_3_ were measured in light and the results were shown in Figure 8d,e. The CB of Cu_2_O and WO_3_ were composed of Cu and W orbitals, respectively. Four peaks of W 4f in light shifted to a higher binding energy compared with these in the dark. Simultaneously, two peaks of Cu 2p in light reversely shifted. It was implied that the photo-induced electrons transferred from WO_3_ component to Cu_2_O component in this heterojunction [50,51]. Specifically, the photoexcited holes in the VB of Cu_2_O would transfer to WO_3_ and recombine with the photoexcited electrons in the CB of WO_3_. The CB potential of Cu_2_O and VB potential of WO_3_ thermodynamically realized photocatalytic CO_2_ reduction and H_2_O oxidation, respectively, and the metallic Cu co-catalyst facilitated the reduction process dynamically.

## 4. Conclusions

In summary, Cu and Cu/Cu_2_O species were synthesized on the hexagonal WO_3_ nanoflake films by the one-step electrodeposition method for photocatalytic CO_2_ reduction with water vapor. The obtained Cu/Cu_2_O/WO_3_ catalyst exhibited excellent photocatalytic performance under visible light irradiation (λ > 400 nm) due to the construction of heterojunction, and the CH_4_, H_2_ and O_2_ yields reached 2.43, 0.32 and 3.45 mmol/g_cat_ after 24 h of illumination, respectively. Notably, CH_4_ molecules were generated as the major product over the Cu/Cu_2_O/WO_3_ catalyst, whereas Cu_2_O/WO_3_ facilitated CO generation. Efficient CH_4_ formation for the Cu/Cu_2_O/WO_3_ catalyst was attributed to the modification of Cu nanoparticles favoring electron–proton transfer from CO to CH_4_. The decreased photocatalytic activity in the cycling experiment was recovered by the regenerated treatment via electro reduction, removing the superfluous W(V) in the composite. Additionally, the S-scheme charge transfer mode and potential mechanism of CO_2_ reduction were proposed by the results of XPS measurement and photocatalytic performance under light illumination with a specific wavelength. The present research may provide a promising strategy to design ternary nanocomposite VLD photocatalysts and inspire further interest in tuning product selectivity for photocatalytic CO_2_ conversion.

## Figures and Tables

**Figure 1 nanomaterials-12-02284-f001:**
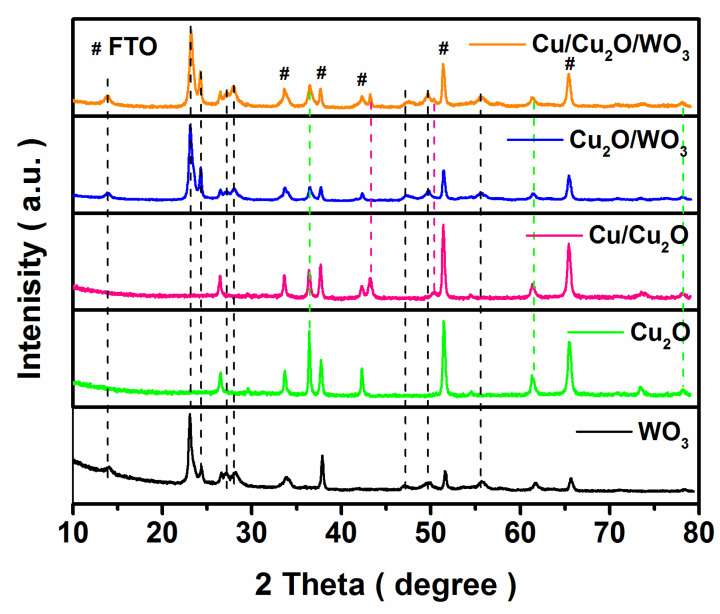
XRD patterns of the bare and composite samples (# sign: the characteristic peak of FTO substrate).

**Figure 2 nanomaterials-12-02284-f002:**
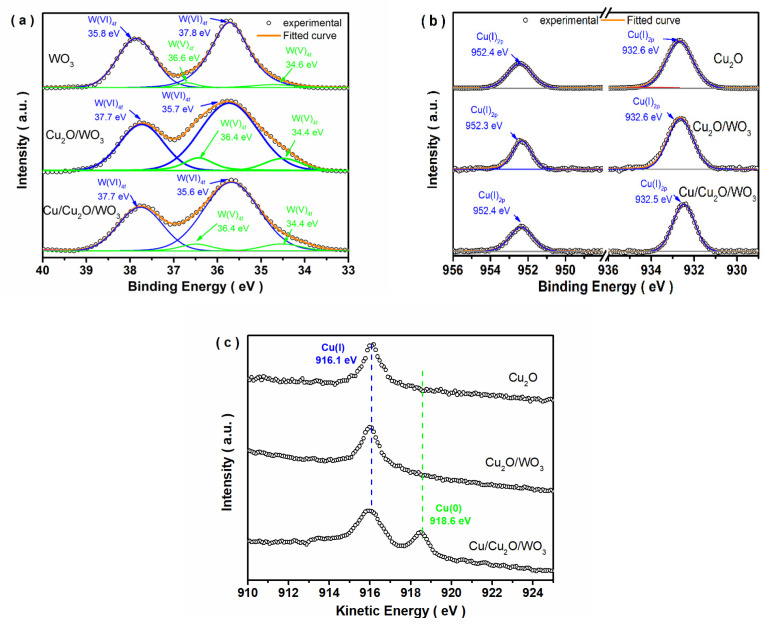
XPS spectra (**a**): W 4f, (**b**): Cu 2p, and Cu LMM auger spectra (**c**) of the obtained samples.

**Figure 3 nanomaterials-12-02284-f003:**
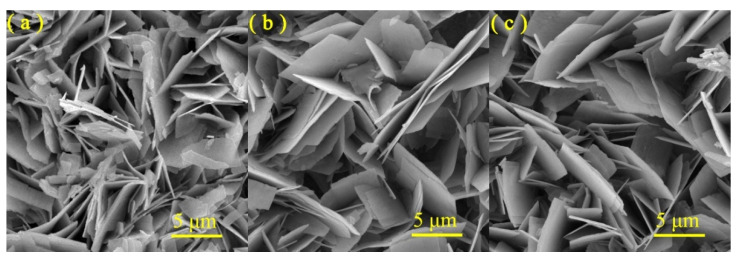
SEM images of WO_3_ (**a**), Cu_2_O/WO_3_ (**b**) and Cu/Cu_2_O/WO_3_ (**c**) samples.

**Figure 4 nanomaterials-12-02284-f004:**
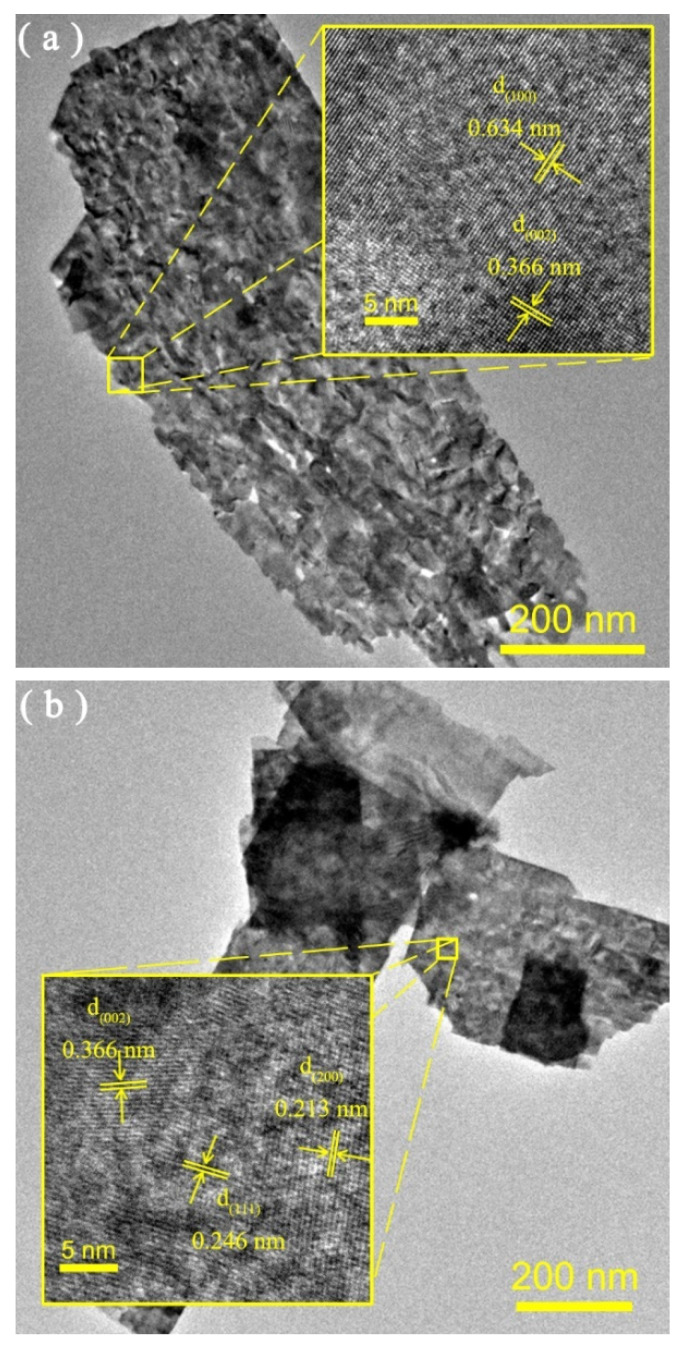
TEM images of WO_3_ (**a**), Cu_2_O/WO_3_ (**b**) and Cu/Cu_2_O/WO_3_ (**c**), inset: HRTEM images), and elemental mapping images of Cu/Cu_2_O/WO_3_ (**d**).

**Figure 5 nanomaterials-12-02284-f005:**
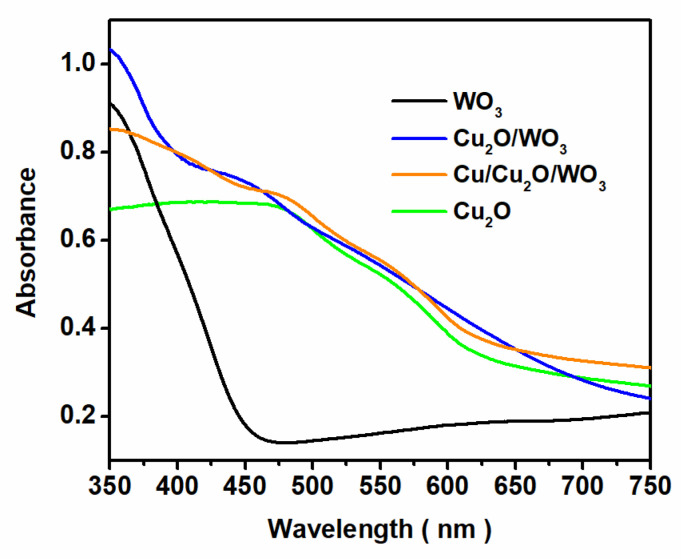
UV-Vis diffuse reflectance spectra of WO_3_, Cu_2_O, Cu_2_O/WO_3_ and Cu/Cu_2_O/WO_3_ composites.

**Figure 6 nanomaterials-12-02284-f006:**
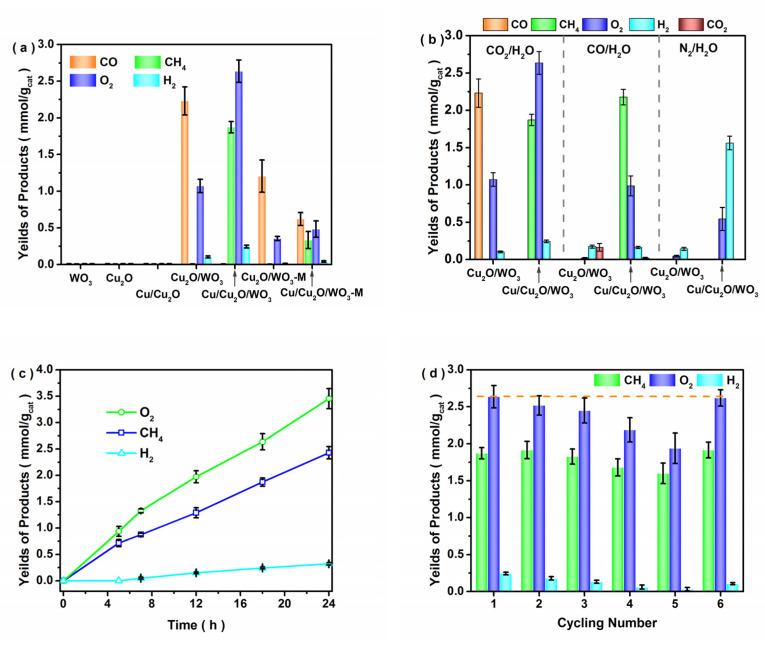
Photocatalytic performances of CO_2_ reduction with water vapor after 18 h of visible light irradiation (>400 nm) over different samples (**a**), photocatalytic activities under different atmospheres (**b**), the yields of products for the Cu/Cu_2_O/WO_3_ composite in the CO_2_/H_2_O atmosphere within 24 h of irradiation (**c**) and product yields for the Cu/Cu_2_O/WO_3_ catalyst in cycling experiment under visible light irradiation for 18 h (**d**).

**Figure 7 nanomaterials-12-02284-f007:**
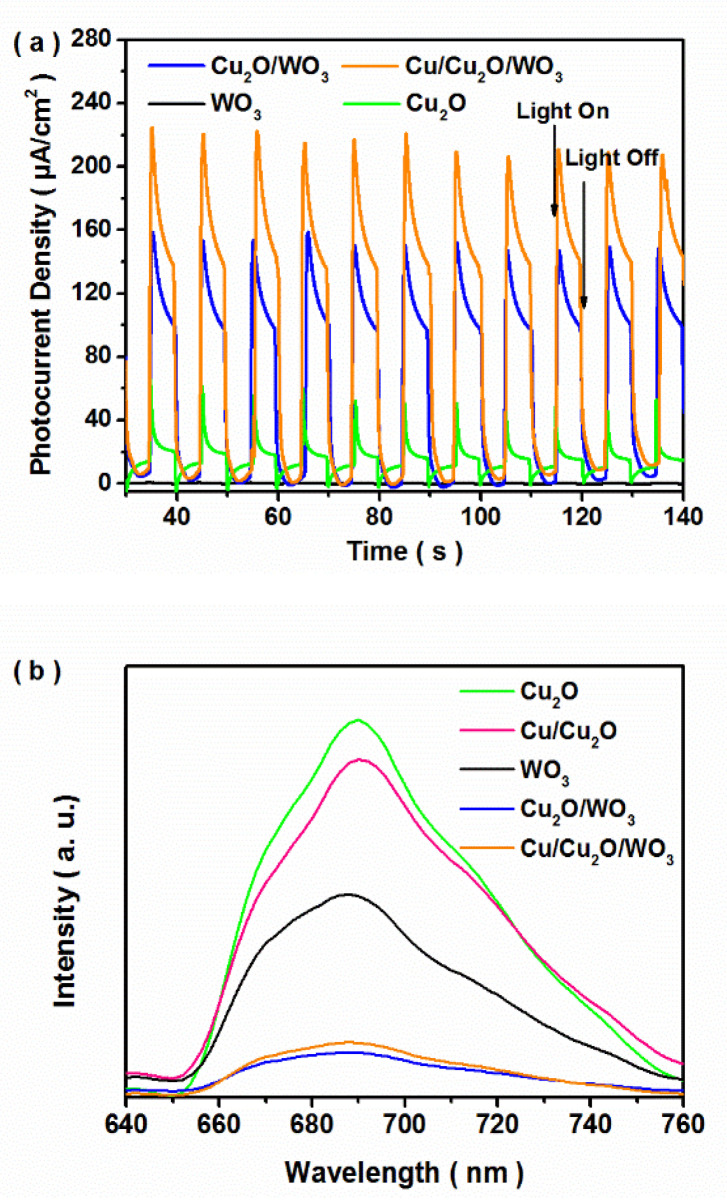
Photocurrent density (**a**) and photoluminescence spectra (**b**) of the obtained samples.

**Figure 8 nanomaterials-12-02284-f008:**
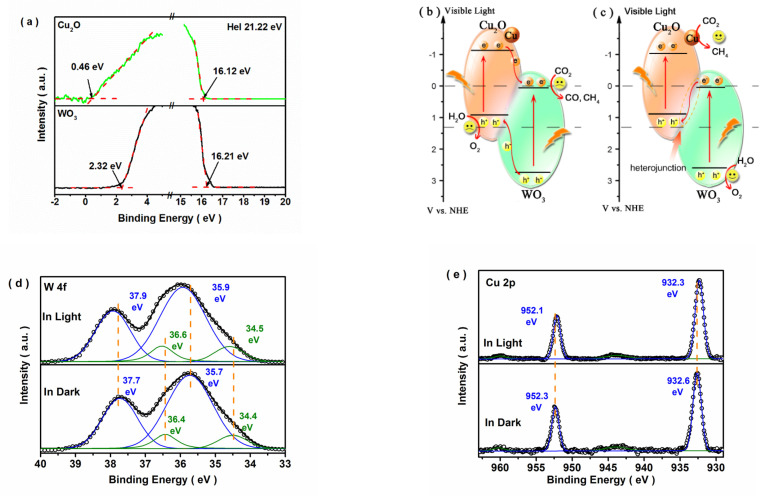
Ultraviolet photoelectron spectra of Cu_2_O and WO_3_ samples measured at the basics of −8 V (**a**), photocatalytic electron transfer modes of the Cu/Cu_2_O/WO_3_ composite: the traditional double transfer mode (**b**) and the S-scheme charge transfer mode (**c**), and XPS spectra of W 4f (**d**) and Cu 2p (**e**) for Cu_2_O/WO_3_ heterojunction in dark/light.

## Data Availability

Data are contained within the article.

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
