# Peer review of "Cu Nanoparticles Modified Step-Scheme Cu2O/WO3 Heterojunction Nanoflakes for Visible-Light-Driven Conversion of CO2 to CH4"

_nanomaterials, 2022, doi:10.3390/nano12132284_

Round 1

Reviewer 1 Report

The authors report the preparation of heterostructured photocatalysts associating Cu2O and WO3 or Cu(0), Cu2O and WO3 by electro-deposition. These catalysts were used for the photo-reduction of CO2 into CH4. Results are of interest and the work is relatively clearly presented. The following comments should be considered by the authors :

- in some parts of the manuscript, the language must be improved (see for example, the first sentence of the abstract). The authors must carefully check the manuscript.

- experimental : 1) clarify the synthetic protocol and specify all amounts of reagents used. 2) if possible , provide the light intensity used during the photocatalytic tests.

- indicate clearly in the text that Cu(0) is formed during the electro-deposition experiment. How was the Cu(0)/Cu2O ratio controlled ? Can the authors comment on the stability of the Cu(0) phase if the catalyst is exposed to air ?

- the authors must determine the Cu/W atomic ratio via ICP analyses for Cu2O/WO3 and Cu/Cu2O/WO3 catalysts.

- figure 5 : remove "a.u." from the y axis (the absorbance is without unit).

- the authors must compare the performance of the Cu/Cu2O/WO3 catalyst for CO2 reduction into CH4 to other heterostructured photocatalysts described in the literature and highlight the advances made.

Author Response

The authors report the preparation of heterostructured photocatalysts associating Cu2O and WO3 or Cu(0), Cu2O and WO3 by electro-deposition. These catalysts were used for the photo-reduction of CO2 into CH4. Results are of interest and the work is relatively clearly presented. The following comments should be considered by the authors :

  1. in some parts of the manuscript, the language must be improved (see for example, the first sentence of the abstract). The authors must carefully check the manuscript.

Author response 1: Thank the reviewer for this view. We have checked the manuscript and some mistakes have been modified in the revised manuscript.

  1. experimental : 1) clarify the synthetic protocol and specify all amounts of reagents used. 2) if possible , provide the light intensity used during the photocatalytic tests.

Author response 2: Thank the reviewer for the constructive suggestions. The amounts of the used reagents and light intensity during the photocatalytic tests (260 mW/cm2) have been supplemented in the revised manuscript.

  1. indicate clearly in the text that Cu(0) is formed during the electro-deposition experiment. How was the Cu(0)/Cu2O ratio controlled ? Can the authors comment on the stability of the Cu(0) phase if the catalyst is exposed to air ?

Author response 3: Thank the reviewer for this comment. We have checked the relevant data and supplemented some experiments.

The content of Cu(0) would increase with the voltage decreasing, when the same electro-deposition cycling was kept. The obtained sample was usually kept in the glass desiccator filled with N2 atomsphere. The Cu/Cu2O/WO3 composite exposure to air for 72 h exhibited similar yields and selectivity of CO and CH4 products (1.87 and 0.0071 mmol/gcat) compared with that of the sample (1.89 and 0.0065 mmol/gcat) kept in the glass desiccator. Meanwhile, there has no obvisous difference between the above two samples (Figure R1), indicating its relative stability when exposed to air.

Figure R1. XRD pattern of the freshed Cu/Cu2O/WO3 sample and the processed Cu/Cu2O/WO3 sample.

  1. the authors must determine the Cu/W atomic ratio via ICP analyses for Cu2O/WO3and Cu/Cu2O/WO3 catalysts.

Author response 4: Thank the reviewer for this comment. The Cu/W atomic ratios for Cu2O/WO3 and Cu/Cu2O/WO3 via ICP analyses were severally calculated to be 16.66% and 18.22% , which have been supplemented in the revised supplementary material.

  1. figure 5 : remove "a.u." from the y axis (the absorbance is without unit).

Author response 5: Thank the reviewer for this suggestion. “a.u.” has been removed from the y axis in Figure 5 in the revised manuscript.

  1. the authors must compare the performance of the Cu/Cu2O/WO3catalyst for CO2 reduction into CH4 to other heterostructured photocatalysts described in the literature and highlight the advances made.

Author response 6: Thank the reviewer for this constructive view. Relevant discussion has been modified in the revised manuscript.

Photocatalytic performances of CuOx based materials for CO2 reduction have been summarized in Table R1 based on the previous studies. The maximum rate of CH4 product over Cu/Cu2O/WO3 was obviously higher than those of the reported catalysts, and high product selectivity was also obtained.

Table R1. Performance comparison of CO2 conversion for the obtained Cu/Cu2O/WO3 catalyst with other reported catalysts

Catalyst

Experimental Conduction

Performance

Reference

Cu/Cu2O/g-C3N4

300W Xe lamp

CO: 10.8 μmol/gcat/h

CH4: 3.1μmol/gcat/h

CH4/CO: 0.29

Carbon, 2022, 193, 272-284.

Cu/CuOx/TiO2

56 mW/cm2 LED (425 nm)

CO: 0.7μmol/gcat/h

CH4: 1.2 μmol/gcat/h

CH4/CO: 1.71

Nanomaterials, 2022, 12, 1584.

Cu-Ti3C2Tx/g-C3N4

300W Xe lamp (170 mW/cm2, 400nm cutoff filter)

CO: 1225.5μmol/gcat/h

CH4: 90 μmol/gcat/h

CH4/CO: 0.07

Chem. Eng. J., 2022, 446, 137028.

1%CuOx/TiO2(101)

300 W Xe lamp

CO: 0.32μmol/gcat/h

CH4: 2.29 μmol/gcat/h

CH4/CO: 7.15

H2: 2.96μmol/gcat/h

Appl. Surf. Sci., 2021, 564, 150407.

Cu2O/Ti3C2Tx

300 W Xe lamp

CO: 17.55 μmol/gcat/h

CH4: 0.96 μmol/gcat/h

CH4/CO: 0.05

Appl. Surf. Sci., 2021, 542, 148685.

Cu2O@Cu3(BTC)2

500 W Xe lamp (400nm cutoff filter), Thin film on copper mesh

CO: 0.73 μ mol for 8h

Angew. Chem. Int. Ed., 2021, 6, 8455-8459.

Cu/Cu2O/WO3

300 W Xe lamp (170 mW/cm2, 400nm cutoff filter, )

After 24 h illumination

CH4: 101.2 μmol/gcat/h

H2: 13.3 μmol/gcat/h

O2: 143.7 μmol/gcat/h

This work

Reviewer 2 Report

The manuscript titled “Cu Nanoparticles Modified Step-Scheme Cu2O/WO3 Heterojunction Nanoflakes for Visible-Light-Driven Conversion of CO2 to CH4” presented by Weina Shi, Ji-Chao Wang, Aimin Chen, Xin Xu, Shuai Wang, Renlong Li, Wanqing Zhang, Yuxia Hou deals with the W based photocatalysts modified by copper species for reduction of CO2. However, there are issues that could be clarified or took into account:

  • Line 16: It looks that some words are missed between “fabricated” and “that”.
  • Line 25: The use of incorrect term “the situ-XPS measurement” instead of “the in situ XPS measurements”.
  • Section 2.2 should content the description of used methods (not in the SI). But I should comment that the copy-paste of SI is not accepted. The description of methods should content: for XRD (source, scanning rate, etc.), for TEM (resolution, acceleration voltage, etc.), for XPS (source, charge correction, background subtraction, software, etc.).
  • Figure 1: the size of the figure should be increased. The all reflection should be marked.
  • The presentation of XPS data should be corrected. 1. The presentation of XPS spectra is incorrect (wrong direction of x-axis). 2. The presentation of Auger spectrum is incorrect (x-axis is scaled in kinetic energy). 3. The Auger-parameter α should be calculated for all Cu2p spectra and the citing literature is necessary. 4. The surface ratio of [Cu]/[W] is demanded and compared with ICP-AES analysis.
  • TEM analysis: It is difficult to understand for which sample Fig. S1 and S2 were done. Fig. S2 looks that the mean size is 4.7±1 nm.
  • The scale in Fig. 5 should be presented (y-axis).
  • The photocatalytic data: the use of Xenon lamp is the old-style approach, now the LED or artificial Sun source are used. For Xe lamp it is not possible to calculate the quantum efficiency, but it is necessary to compare the photocatalytic systems. The reduction rate of CO2 is not calculated that does not allow comparison the Cu-W systems with the other ones (Cu/TiO2, C3N4, etc). The information about CO2 reduction rate should be given in terms of µmol/min or µmol/(g*h).
  • Lines 260-261: CO could not be an intermediate, CO is the final product of CO2 reduction. The mechanism of reduction of CO2 to CH4 does not include the CO-intermediate.

The choice of methods to study the catalysts looks well. The manuscript demands some corrections, and after correction stage it  could be accepted for publication.

Author Response

The manuscript titled “Cu Nanoparticles Modified Step-Scheme Cu2O/WO3 Heterojunction Nanoflakes for Visible-Light-Driven Conversion of CO2 to CH4” presented by Weina Shi, Ji-Chao Wang, Aimin Chen, Xin Xu, Shuai Wang, Renlong Li, Wanqing Zhang, Yuxia Hou deals with the W based photocatalysts modified by copper species for reduction of CO2. However, there are issues that could be clarified or took into account:

  1. Line 16: It looks that some words are missed between “fabricated” and “that”.

Author response 1: Thank the reviewer for this view. We have carefully checked the manuscript and some mistakes have been modified.

  1. Line 25: The use of incorrect term “the situ-XPS measurement” instead of “the in situ XPS measurements”.

Author response 2: Thank the reviewer for this comment. We have checked the manuscript, and the correct description of “the in situ XPS measurements” has been added into the revised manuscript.

  1. Section 2.2 should content the description of used methods (not in the SI). But I should comment that the copy-paste of SI is not accepted. The description of methods should content: for XRD (source, scanning rate, etc.), for TEM (resolution, acceleration voltage, etc.), for XPS (source, charge correction, background subtraction, software, etc.).

Author response 3: Thank the reviewer for the constructive opinions. The detailed test conditions of the used methods have been added into section 2.2 in the revised manuscript.

The crystal phases of the samples were recorded using X−ray diffractometer (PANalytical X' pert PRO, Netherlands) with Cu Kα irradiation source (λ= 0.154 nm) and 0.15o/s scanning step. A scanning electron microscope (SEM, Nova NanoSEM 450, FEI) using the acceleration 300 kV voltage was used to characterize the morphology of the obtained products. Transmission electron microscopy (TEM) was obtained on a Tecnai G2 F20 S-TWIN electron microscope. Further, high-resolution transmission electron microscopy (HRTEM) and Energy Dispersive X-Ray spectroscopy (EDX) was employed and the corresponding fast Fourier transform (FFT) was evaluated by Gatan Digital Micro-graph software (Gatan Inc., America). X-ray photoelectron spectroscopy (XPS) measurements were carried out at room temperature on a Thermo escalab 250Xi X-ray Photoelectron Spectrometer with a monochromatic Al Kα radiation(hv = 1486.6 eV). For XPS analysis, the samples without exposing to air are dried in N2 flow gas and vacuum packed to avoid any impurity. All spectra were calibrated to the C1s peak at 284.6 eV. The peak position was estimated using a fitting procedure based on summation of Lorentzian and Gaussian functions using the XPSPEAK 4.1 program. UV-Vis diffuse reflectance spectra (DRS) were performed on a scan UV-Vis spectrometer (Cary 5000). The composition for the composites was determined by ICP-AES analysis using Thermo Scientic iCAP 6000 spectrometry. Photoelectrochemical test was recorded in a conventional three-electrode system by a CHI 660E electrochemical workstation (Chenhua, Shanghai, China). The photo-currents of the photocatalysts were measured at 0.0 V (vs. Ag/AgCl) in Na2SO4 aqueous solution after being purged by N2 under UV-visible light with AM 1.5G filter.

  1. Figure 1: the size of the figure should be increased. The all reflection should be marked.

Author response 4: Thank the reviewer for the constructive suggestions. The size of Figure 1 has been increased, and the characteristic peaks of Cu, Cu2O and WO3 have been severally marked with red , green and black dotted lines in the revised manuscript.

Figure R1. XRD patterns of the bare and composite samples.

  1. The presentation of XPS data should be corrected. 1. The presentation of XPS spectra is incorrect (wrong direction of x-axis). 2. The presentation of Auger spectrum is incorrect (x-axis is scaled in kinetic energy). 3. The Auger-parameter α should be calculated for all Cu2p spectra and the citing literature is necessary. 4. The surface ratio of [Cu]/[W] is demanded and compared with ICP-AES analysis.

Author response 5: Thank the reviewer for the constructive comments. The correct presentation of XPS spectra (right direction of x-axis) and Auger spectrum (x-axis is scaled in kinetic energy) have been added into the revised manuscript according to the constructive suggestions. The Auger parameter can be calculated from the equation of α¢ = Ek (Auger electron) + Eb (photoelectron). Here, Ek is the kinetic energy, and Eb is the binding energy. The Auger parameter value of the Cu(I) and Cu(0) in the Cu/Cu2O/WO3 composite sample were determined to be 1848.7 eV and 1851.2 eV, respectively. Y. Tong et al. proven that the Auger parameter value of the Cu2O nanoparticles was determined to be 1848.7 eV.( Electrochim. Acta, 2012, 62, 1-7) G. Moertti et al. also reported that the Auger parameters of Cu2O (Cu+) and Cu(0) were calculated to be 1849.17 and 1851.24 eV eV, respectively.(Surf. Interface Anal. 2022, 54, 803-812) In this work, the relevant results agreed with the above studies. Additionally, the ratio of Cu/W in the composite could be calculated by the following formula:

nCu/nW = (ICu/SCu) / (Iw/Sw)

where n, I and S were on behalf of atomic number, peak area and sensitivity factor of each element, respectively. The results of Cu/W ratios were summarized in Table R1, and the ratios calculated by XPS were slightly higher than those of ICP-AES measurement. It was because that XPS measurement e is a technique for analyzing the material’s surface , and the Cu/Cu2O and Cu2O mainly existed on the surface of WO3 nanoflake by electro-deposition method. While, the ICP-AES results were measured by completely dissolving the composite, and thus the Cu/W ratios was lower than that of XPS. aHowever, there was little difference of Cu/W ratios between Cu/Cu2O/WO3 and Cu2O/WO3 samples. The above results and literature have been added into the revised manuscript.

Table R1. ICP-AES and XPS analysis of Cu2O/WO3 and Cu/Cu2O/WO3 samples.

Sample

Ratio of Cu/W atoms (%)

ICP-AES

XPS

Cu2O/WO3

16.66

19.50

Cu/Cu2O/WO3

18.22

20.01

  1. TEM analysis: It is difficult to understand for which sample Fig. S1 and S2 were done. Fig. S2 looks that the mean size is 4.7±1 nm.

Author response 6: Thank the reviewer for this constructive comment. The mean size of Cu nanoparticle in Fig. S2 has been remeasured to be 5.6±1.1 nm. Cu nanoparticles were scattered around the Cu/Cu2O/WO3 nanoflake by ultrasonic exfoliation, and the size distribution of Cu nanoparticles in the selected area in Figure 4c (framed in dark red, corresponding enlarged view in Figure S1) obeyed a logical normal distribution with an average diameter of 5.6±1.1 nm (Figure S2). The above related description has been added into the revised manuscript.

  1. The scale in Fig. 5 should be presented (y-axis).

Author response 7: Thank the reviewer for this suggestion. The corresponding scale has been added into Figure 5 in the revised manuscript.

  1. The photocatalytic data: the use of Xenon lamp is the old-style approach, now the LED or artificial Sun source are used. For Xe lamp it is not possible to calculate the quantum efficiency, but it is necessary to compare the photocatalytic systems. The reduction rate of CO2 is not calculated that does not allow comparison the Cu-W systems with the other ones (Cu/TiO2, C3N4, etc). The information about CO2 reduction rate should be given in terms of µmol/min or µmol/(g*h).

Author response 8: Thank the reviewer for the constructive suggestions. The CO2 reduction rates have been given in terms of mmol/gcat/h. In this study, the light intensity of Xenon lamp was 280 mW/cm2 for CO2 reduction. Photocatalytic performances of CuOx based materials for CO2 reduction have been summarized in Table R2 based on the previous studies. The maximum rate of CH4 product over Cu/Cu2O/WO3 was obviously higher than those of the reported catalysts, and high product selectivity was also obtained.

Table R2. Performance comparison of CO2 conversion for the obtained Cu/Cu2O/WO3 catalyst with other reported catalysts

Catalyst

Experimental Conduction

Performance

Reference

Cu/Cu2O/g-C3N4

300W Xe lamp

CO: 10.8 μmol/gcat/h

CH4: 3.1μmol/gcat/h

CH4/CO: 0.29

Carbon, 2022, 193, 272-284.

Cu/CuOx/TiO2

56 mW/cm2 LED (425 nm)

CO: 0.7μmol/gcat/h

CH4: 1.2 μmol/gcat/h

CH4/CO: 1.71

Nanomaterials, 2022, 12, 1584.

Cu-Ti3C2Tx/g-C3N4

300W Xe lamp (170 mW/cm2, 400nm cutoff filter)

CO: 1225.5μmol/gcat/h

CH4: 90 μmol/gcat/h

CH4/CO: 0.07

Chem. Eng. J., 2022, 446, 137028.

1%CuOx/TiO2(101)

300 W Xe lamp

CO: 0.32μmol/gcat/h

CH4: 2.29 μmol/gcat/h

CH4/CO: 7.15

H2: 2.96μmol/gcat/h

Appl. Surf. Sci., 2021, 564, 150407.

Cu2O/Ti3C2Tx

300 W Xe lamp

CO: 17.55 μmol/gcat/h

CH4: 0.96 μmol/gcat/h

CH4/CO: 0.05

Appl. Surf. Sci., 2021, 542, 148685.

Cu2O@Cu3(BTC)2

500 W Xe lamp (400nm cutoff filter), Thin film on copper mesh

CO: 0.73 μ mol for 8h

Angew. Chem. Int. Ed., 2021, 6, 8455-8459.

Cu/Cu2O/WO3

300 W Xe lamp (170 mW/cm2, 400nm cutoff filter, )

After 24 h illumination

CH4: 101.2 μmol/gcat/h

H2: 13.3 μmol/gcat/h

O2: 143.7 μmol/gcat/h

This work

  1. Lines 260-261: CO could not be an intermediate, CO is the final product of CO2 reduction. The mechanism of reduction of CO2 to CH4 does not include the CO-intermediate.

Author response 9: According to the constructive opinions, the mechanism of reduction of CO2 to CH4 has been modified in the revised manuscript.

In the CO/H2O atmosphere, more CH4 molecules were produced over Cu/Cu2O/WO3 compared with that of Cu2O/WO3. Meanwhile, In the N2/H2O atmosphere, the H2 yield from water splitting was improved with the existence of Cu in the composite. It was indicated that Cu nanoparticles promoted the electron transfer and proton aggregation on the catalyst surface, improving the hydrogenated process and C=O breakage for CO2 reduction over the Cu/Cu2O/WO3 catalyst.

The choice of methods to study the catalysts looks well. The manuscript demands some corrections, and after correction stage it could be accepted for publication.

Round 2

Reviewer 1 Report

All corrections were made by the authors. The manuscript can be accepted by Nanomaterials.

Reviewer 2 Report

The authors presented the revised version of manuscript. They took into account all comments and remarks, answered all questions. The current version of manuscript could be accepted for publication.